# Do Nurses Thrive in Their Organization? Validation of the Short Form of Nurses’ Organizational Health Questionnaire

**DOI:** 10.3390/nursrep15120432

**Published:** 2025-12-05

**Authors:** Alessandro Sili, Maddalena De Maria, Valerio Della Bella, Jacopo Fiorini, Claudio Barbaranelli

**Affiliations:** 1Nursing Department, Tor Vergata University Hospital, 00133 Rome, Italy; alessandro.sili@ptvonline.it (A.S.); jacopo.fiorini@ptvonline.it (J.F.); 2Department of Life Science, Health, and Health Professions, Link Campus University, 00165 Rome, Italy; m.demaria@unilink.it; 3Italian Evidence-Based Practice Nursing Scholarship, A JBI Centre of Excellence, 00136 Rome, Italy; 4Department of Psychology, Faculty of Medicine and Psychology, Sapienza University of Rome, 00185 Rome, Italy; claudio.barbaranelli@uniroma1.it

**Keywords:** nurses, occupational health, organizational culture, working conditions, surveys and questionnaires, psychometrics

## Abstract

**Background/Aim:** The quality of care provided to patients was closely related to the nursing staff’s well-being and their experience within the organization. This study aimed to evaluate the psychometric properties of the short form of the Nurses’ Organizational Health Questionnaire (QISO-SF), with a focus on its relevance for assessing nurses’ organizational well-being in healthcare environments. The study examined the instrument’s structural validity and internal consistency. **Methods:** A secondary analysis was conducted using data from three cross-sectional studies, including 1279 nurses providing direct patient care across various Italian healthcare settings. Dimensionality of the QISO-SF was tested via confirmatory factor analysis (CFA), and reliability was assessed using ordinal omega coefficients (ω). **Results:** The QISO-SF comprises 48 items across 11 dimensions, grouped into 5 scales: Comfort, Organizational Context and Relational Processes, Workload, Positive and Negative Indicators, and Psychophysical Distress. The instrument demonstrated good structural validity (RMSEA = 0.048–0.094; CFI = 0.967–0.994) and satisfactory reliability (ω = 0.644–0.857). By maintaining the theoretical framework of the original questionnaire while reducing completion time, the short form is suitable for evaluating nurses’ work-related quality of life and organizational well-being. **Conclusions:** The QISO-SF is a concise, reliable, and valid tool to assess work-related quality of life and Organizational health in nursing professionals. Its use can support interventions aimed at promoting well-being in healthcare settings.

## 1. Introduction

The quality of care provided to patients was closely related to the nursing staff’s well-being and their experience within the organization. Nurses who had good physical, psychological, and social health demonstrated a greater sense of belonging to their profession and organization [1], performed better [2], and delivered high-quality and safe care [3]. In the nursing work environment, the state of well-being generated by the organization was defined as Nursing Organizational Well-being, meaning “the result of nurses’ perceptions of their work environment, influenced by the organization’s ability to promote healthy working conditions. Positive interpersonal relationships in the professional team and the interaction between nursing work demands and the available nursing resources shaped this well-being” [4].

In recent years, numerous studies have investigated the work environment conditions that could produce a state of organizational well-being in nurses [1]. The study of the characteristics of nursing settings, measured using instruments that were easy to use and proven valid and reliable, enabled the identification of factors that could promote or undermine nursing organizational well-being. The well-being of professionals could consequently influence outcomes for nurses (satisfaction and well-being), patients (nursing-sensitive outcomes), and organizations (absenteeism and performance) [2]. However, the determinants of a healthy work environment were multiple and concerned not only the physical work characteristics (suitable environments), but also the social (e.g., interpersonal relationships) and psychological (e.g., perceived workloads) spheres. This complexity made it difficult to study this concept [1].

### Background

A recent literature review identified a wide variety of instruments for measuring organizational well-being in nursing [5], highlighting significant heterogeneity in explored dimensions (from emotional state to work demands and resources) [6,7] and related items [6]. Due to this multidimensionality, it was difficult to find an instrument that simultaneously considered all dimensions related to nurses’ organizational well-being. This variability created a fragmented landscape of instruments, making it difficult for researchers to identify a comprehensive measure of nursing organizational well-being and limiting the ability to compare or synthesize findings across studies [8].

The literature review by Gioiello et al. [5] revealed that the Nursing Questionnaire on Organizational Health (QISO) [9] showed good psychometric properties in terms of validity and reliability, allowing the measurement of nursing organizational well-being through its 8 scales, 19 dimensions, and 118 items. Using this structure, the QISO enabled the study of the nursing work environment through eight domains, including satisfaction with the physical environment, organizational structure, relationships, and stress factors. The assessment of well-being through the QISO also involved the perception of the work environment’s safety and the organization’s openness to innovation. It also included positive and negative indicators related to behaviors and attitudes that nurses might exhibit when expressing their work satisfaction or dissatisfaction.

The QISO was validated on 1279 nurses in different clinical settings such as surgery, medicine, emergency departments, operating theaters, and the critical care area and demonstrated good psychometric properties. In developing the QISO, a group of six experts, including nursing managers, psychometricians, university professors, and experts on the topic of “Organizational well-being”, assessed the relevance and effectiveness of each item for measuring nursing organizational well-being. This process resulted in the modification of certain items and the addition of new ones. The scales’ dimensionality was tested through exploratory factor analysis, using principal axis factorization method. The internal consistency of each scale from the instrument was tested using Cronbach’s alpha, which ranged from 0.72 to 0.90 across all scales.

Despite its strengths, the QISO exhibited several limitations in its long-term use. Specifically, the number of items led several researchers to use only some of the instrument’s subscales, such as job satisfaction and interpersonal relationships with colleagues or head nurse [10]. While this partial use reduced the compilation time, it compromised the loss of a global understanding of the aspects that contribute to organizational well-being in nursing, which the full instrument was designed to provide [11].

Organizational well-being is a multidimensional construct influenced by several aspects of nursing work [12]. Demands and resources do not act in isolation but emerge from the dynamic interaction between the social climate, work demands, and nurses’ perceptions of their organizational experience [13]. Therefore, it was necessary to employ instruments that captured this complexity without compromising the validity and reliability of the data [14]. Partial measurement of a construct could introduce a risk of measurement error, potentially distorting the study of the relationships between the variables [15] and thus compromising research quality [16]. The need for concise, yet complete, versions of already validated questionnaires has emerged as a recurring theme in the literature. Shorter instruments would reduce responders’ compilation time and study costs, improve the quality and quantity of responses provided, and significantly reduce the respondent burden [14,15].

Considering the limitations highlighted, an instrument that considered the entire nursing work context, but with several items that did not affect the quality of the collected data, was needed. The domains measured by the QISO analyzed the entire work context, but, to date, a valid and reliable short version of the instrument is not available. This lack implied a delay in any screening process for organizational well-being and the subsequent design of tailored interventions to improve it [17]. Therefore, testing the psychometric properties of a short form of the QISO-SF became necessary to measure and monitor the characteristics of nursing work contexts, analyze their impact on work and health outcomes for staff and patients, and contribute to improving the quality of care delivered. Considering these knowledge gaps, this study aimed to test the psychometric properties (structural validity and internal consistency reliability) of the QISO-SF in a sample of nurses.

## 2. Materials and Methods

### 2.1. Design and Datasets

To validate the QISO-SF, a secondary analysis of data from three previous studies was conducted. Data were drawn from three studies conducted between 2010 and 2011 in comparable clinical contexts. The first and largest dataset included nurses from multiple inpatient settings (medical, surgical, operating rooms, and emergency units) [9]. The second dataset consisted primarily of nurses working in emergency departments, while the third dataset mainly included nurses from operating rooms and medical wards [18,19]. These datasets comprised nurses operating within comparable organizational contexts and clinical environments, and were collected using consistent procedures and the same instrument. Such conceptual and methodological homogeneity supports their combination for psychometric validation purposes [11].

### 2.2. Participants

All participants were nurses employed in direct patient care, belonging to similar inpatient areas (e.g., medical area, surgical area, emergency area), with the same contract types (e.g., full-time, part-time) and working shifts (e.g., day only, rotating day and night shifts). Data were collected in facilities sited to the Italian National Health System.

### 2.3. Data Collection

The questionnaires were administered in paper format and consisted of a section dedicated to the collection of socio-demographic and occupational characteristics and another section including the extended version of the QISO organized in the eight scales (118 items). A single researcher, who explained the objectives of the research, the voluntary nature of participation and the guarantee of anonymity enrolled the participants. The average response time of the questionnaire was estimated to be approximately 20 min.

### 2.4. Ethical Considerations

The study was approved by the Ethics Committee of the University Hospital of Rome Tor Vergata (No. 172.24 of 26 September 2024) and adhered to the principles outlined in the Declaration of Helsinki [20]. Each participant expressed a willingness to participate in the study through written informed consent. Confidentiality and anonymity were guaranteed by assigning a numerical code to each participant.

### 2.5. Instruments

The Nursing Questionnaire for Organizational Health (QISO) consisted of 118 items grouped into eight subscales exploring the following domains: Comfort, Organizational Context and Relational Processes, Stress Factors, Safety, Task Tolerability, Openness to Innovation, Positive and Negative Indicators of Job Satisfaction, and Psychophysical Distress. The QISO was previously validated on a sample of 1279 nurses and showed good indicators of validity and reliability for each of the subscales, in particular the following: (a) The “Comfort” scale, which assesses the comfort of the working environment through 10 items and uses a 4-step Likert response scale (from 1 = insufficient to 4 = good), showed a unidimensional factorial structure and adequate reliability properties (explained variance (EV) = 42%, α = 0.87, inter-item correlation (IIC) ≥ 0.45). (b) The “Organizational Context and Relational Processes” scale, which assesses the entire organizational context and the relationships established in the workplace through 35 items and using a 4-step Likert response scale (1 = never to 4 = often), showed a 6-factor factorial structure (coordinators’ perception, perception of organizational efficiency, perception of organizational effectiveness, perception of colleagues, perception of competence enhancement and perception of conflict) and adequate reliability properties (EV = 4.2–11.2%, α = 0.62–0.88, IIC ≥ 0.30–0.60). (c) The ‘Stress Factors’ scale, which assesses the level of stress that work activities can generate through 3 items and using a 4-step Likert response scale (1 = not at all to 4 = very much), showed a one-dimensional factorial structure and adequate reliability properties (EV = 5.1%, IIC ≥ 0.20). (d) The “Safety” scale, which assessed the nurses’ perception of the level of safety measured in the different areas using 7 items and a 4-step Likert scale (1 = insufficient to 4 = good), showed a one-dimensional factorial structure and adequate reliability properties (EV = 47%, α = 0.85, r_it ≥ 0.40). (e) The “Tolerability of Tasks” scale assessing the area of less desirable components of work performance through 8 items and using a 4-step Likert scale (from 1 = not at all to 4 = very much) showed a 2-factor factorial structure (work fatigue, integration, and teamwork) and adequate reliability properties (EV = 7.6–19%, α = 0.56–0.77, IIC ≥ 0.30–0.50). (f) The “Openness to Innovation” scale, assessing the participants’ perception of their company’s attention to innovations using 9 items and a 4-step Likert scale (1 = never to 4 = often), showed a one-dimensional factorial structure and adequate reliability properties (EV = 58%, α = 0.92, IIC ≥ 0.65). (g) The “Positive and Negative Indicators” scale, assessing nurses’ satisfaction with their organization using 31 items and using a 4-step Likert scale (1 = never to 4 = often), showed a 4-factor structure (general satisfaction, satisfaction with management, satisfaction with one’s operational unit and dissatisfaction) and adequate reliability properties (EV = 8.1–15.5%, α = 0.81–0.90, IIC ≥ 0.55–0.75). (h) The “Psychophysical distress” scale, which assesses the presence of psychic disturbances in the previous 6 months using 8 items and a 4-step Likert scale (1 = never to 4 = often), showed a one-dimensional factorial structure and adequate reliability properties (EV = 41.6%, α = 0.84, IIC ≥ 0.35).

### 2.6. Item Reduction

Following the indications of Polit and Beck [21], item reduction was carried out by a panel of five experts in nursing organizational well-being, academic faculty, and psychometrics. Using a dichotomous scale (0 = exclude; 1 = include), each expert independently evaluated every item, assigning a score of 0 when the item was considered inappropriate and 1 when it was deemed appropriate and aligned with the original QISO structure. Only items reaching full agreement (5 out of 5) were retained in the final version of the QISO-SF [11].

### 2.7. Statistical Analysis

Data analysis was conducted in five phases. First, the sociodemographic characteristics of the sample were described using means (M) and standard deviations (SD) for continuous variables and frequencies and percentages for categorical variables.

Third, the factorial structure of the QISO-SF was investigated using confirmatory factor analysis (CFA). Based on the validation study in which the factorial structure of the QISO was tested performing an exploratory factor analysis (principal axis factoring, PAF) and according to theoretical reasons supported by authors, CFA was specified to identify a core set of items that would provide the best approximation of a simple congeneric structure of the short form of the QISO [22,23]. Since the theoretical structure of the instrument is already well established in the literature, and because an exploratory factor analysis in this context could have led to an unnecessary re-specification of an established model, the authors chose not to perform an exploratory factor analysis, as recommended in the methodological literature [22,23]. Five CFAs were conducted separately for each scale (Comfort, Organizational Context and Relational Processes, Workload, Positive and Negative Indicators, and Indicators of Psychophysical Distress) of the QISO. Since QISO has ordinal items and normality assumption cannot be assumed, the weighted least squares mean and variance adjusted estimator (WLSMV) was used [22,23,24]. Model fit was assessed with the following fit indices: chi-square (χ^2^), the comparative fit index (CFI; value > 0.95 indicate good fit), the Tucker–Lewis index (TLI; values > 0.95 indicate a good fit), the root mean square error of approximation (RMSEA; values between 0.05 and 0.08 indicate a moderate fit), and the standardized root mean square residual (SRMR; values ≤ 0.08 indicate a good fit) [25]. As the analysis aimed to confirm an established theoretical structure, parsimony and comparison-based indices (such as AIC, PNFI, PGFI, and PRATIO) were not used [22,23]. Chi-square (χ^2^) statistics were reported, but due to the large sample and the sensitivity of the chi-square likelihood ratio test to sample size, chi-square test results were not used in interpreting model fit [26]. Consistent with current guidelines, the χ^2^/df ratio was not reported because its cutoffs are considered arbitrary; therefore, model evaluation relied on RMSEA, CFI, TLI, and SRMR [22]. Factor loadings were considered adequate to confirm the construct if greater than |0.30| [27]. Fourth, internal consistency reliability was determined with ordinal-omega reliability coefficients, considered adequate if > of 0.70 [22,23].

A *p*-value < 0.05 was considered statistically significant. Descriptive statistics were computed with SPSS v.25 [28]. The CFA and MG-CFA were conducted with MPLUS v.8.5 [29] and JAMOVI [30,31] for ordinal-omega (**ω**) reliability coefficients [22,23].

## 3. Results

### 3.1. Characteristics of the Sample

The characteristics of the sample are reported in Table 1. The study enrolled 1279 nurses, predominantly female (*n* = 842, 65.83%), with an average age of 34.47 years (SD = 7.16). Most of the nurses work in the operating room department (*n* = 268, 20.95%), followed by those in the emergency department (*n* = 265, 20.72%) and intensive care departments (*n* = 251, 19.62%)**.**

### 3.2. Dimensionality of QISO-SF Scale

#### 3.2.1. Comfort Scale

Accordingly, with a previous study one factor confirmatory model was specified for the short form of the Comfort scale. The confirmatory factor analysis (CFA) model, wherein the six items were posited as indicators of a distinct factor, showed ambiguous fit indices, pointing to a possible misfit: χ^2^ (9, *n* = 1146) = 297.905, *p* < 0.001; RMSEA = 0.167 (90% CI [0.151, 0.184]) *p*(RMSEA < 0.05) < 0.001; CFI = 0.950; TLI = 0.917; and SRMR= 0.046. An inspection of the modification indices reveals that the misfit originated from excessive covariance between items #6 and #5, as well as between items #1 and #2. According to the recommendations of Fornell & Larcker [32] and Bagozzi [33], covariances between residuals could be specified if they were theoretically justifiable and did not impact parameter estimations, which was the case in our analysis. There were robust methodological reasons supporting these error covariances [34]. All of these covariances were associated with items occupying adjacent positions on the scale. Adjacent pairs of positively worded items might exhibit a correlation pattern that increases with proximity and decreased with greater inter-item distance, commonly referred to as a “proximity” effect [35]. Error covariance was employed to accommodate the additional source of item covariance introduced by item proximity. Furthermore, all these elements referred to environmental factors that could influence the safety and health of nurses working in hospitals. The goodness-of-fit indices of the respecified model were adequate: χ2 (7, *n* = 1146) = 40.696, *p* < 0.001; RMSEA = 0.065 (90% [CI 0.046, 0.085]) *p*(RMSEA < 0.05) = 0.090; CFI = 0.994; TLI = 0.988; and SRMR = 0.017. All factor loadings were significant and ≥0.650 (Table 2).

#### 3.2.2. The Organizational Context and Relational Processes Scale

According to a previous study, a five-factor confirmatory model was specified. Organizational Context and Relational Processes is described as comprising “Perception of head nurses” (F1), “Perception of effectiveness and efficiency” (F2), “Perception of nursing colleagues” (F3), “Perception of rewards” (F4), and “Perception of conflicts” (F5). F1 was measured by items #7, #8, and #9, and #10, F2 by items #11, #12, #13, and #14, F3 by items #15, #16, and #17, F4 by #18 and #19, and F5 by #20, #21, and #22. This model yielded a good fit: χ2 (94, *n* = 1146) = 343.414, *p* < 0.001; RMSEA = 0.048 (90% CI [0.043, 0.054]) *p*(RMSEA < 0.05) = 0.706; CFI = 0.977; TLI = 0.971; and SRMR= 0.037. All factor loadings were significant and ≥0.610 (Table 2).

#### 3.2.3. The Workload Scale

For the workload scale, a one factory confirmatory model was specified that yielded the following fit indices: χ^2^ (9, *n* = 1146) = 144.309, *p* < 0.001; RMSEA = 0.115 (90% CI [0.098, 0.131]) *p*(RMSEA < 0.05) < 0.001; CFI = 0.966; TLI = 0.943; and SRMR= 0.038. An exploration of the modification indices showed that the cause of misfit was in the excessive covariance between items #28 and #27, and #26 and #24. Covariances between residuals could be specified since theoretically justifiable and the fit of the model was the follows: χ^2^ (7, *n* = 1146) = 34.871, *p* < 0.001; RMSEA = 0.059 (90% CI [0.040, 0.079]) *p*(RMSEA < 0.05) *=* 0.20; CFI = 0.993; TLI = 0.985; and SRMR= 0.018. Items #28 and #27 shared a similar semantic structure. Regarding items #26 and #24, the role of a nurse involved a profound interconnection between emotional labor and mental workload. These two job facets were intricately linked, requiring a delicate balance to navigate the demands of both the emotional and cognitive dimensions of their responsibilities. All factor loadings were significant and ≥0.578 (Table 2).

#### 3.2.4. Positive and Negative Indicators Scale

For the Positive and negative indicators scale, a three-factor confirmatory model was specified. The Management Satisfaction (F1) was measured by four items (#31, #32, #33 and #34), the Nursing Caring Satisfaction was measured (F2) by five items (#29, #30, #35, #36 and #37) and Job Dissatisfaction (F3) by six items (#37, #38, #39, #40,). Expectations of the modification indices highlighted that the cause of misfit was in the excessive covariance between items #29 and #30, and #41 and #42. Covariances between residuals, since theoretically justifiable, were specified, and the fit of the model was as follows: χ^2^ (85, *n* = 1146) = 946.148, *p* < 0.001; RMSEA = 0.094 (90% [CI 0.089, 0.099]) *p*(RMSEA < 0.05) <0.001; CFI = 0.967; TLI = 0.959; and SRMR= 0.060. Items #29 and #30, and #41 and #42 therefore share residual variance due to the proximity effect. In addition, the specified correlations between residuals find support in theoretical underpinnings. In the first instance (#29 and #30), a positive desire to engage in work and the perception of personal fulfillment through one’s professional role served as indicators of organizational well-being [36]. Conversely, in the second instance (Items #41, Items #42), the sensation of insignificance within the organizational framework and a perceived lack of adequate recognition contributed to indicators of organizational dissatisfaction [36]. All factor loadings were significant and ≥0.538 (Table 2).

#### 3.2.5. Indicators of Psychophysical Distress Scale

Finally, for the Indicators of Psychophysical Distress scale, a one factor model was specified. Items #43, #44, #45, #46, and #47 measured indicators of psychophysical distress. The goodness-of-fit indices of this model were adequate: χ^2^ (5, *n* = 1146) = 36.533, *p* < 0.001; RMSEA = 0.074 (90% CI [0.053, 0.098]) *p*(RMSEA < 0.05) = 0.033; CFI = 0.989; TLI = 0.977; and SRMR = 0.020. All factor loadings were significant and ≥0.577 (Table 2). Table 3 presents the fit indices for all QISO-SF scales, summarizing the goodness-of-fit statistics for each model.

### 3.3. Reliability of QISO-SF Scale

Coefficient reliability of the QISO-SF scale is reported in Table 4. The reliability of the Comfort scale resulted in the omega coefficient of 0.779. The omega coefficient reliability values were 0.703, 0.748, 0.687, 0.644, and 0.677 for “Perception of head nurses”, “Perception of effectiveness and efficiency”, “Perception of nursing colleagues”, “Perception of rewards”, and “Perception of conflicts” factors, respectively. For the Workload scale, the omega coefficient was 0.810. For the Positive and Negative Indicators scale, the omega coefficients were 0.847, 0.857, and 0.845 for Management Satisfaction, Nurses Caring Satisfaction, and Job Dissatisfaction, respectively. Finally, for the Indicators of Psychophysical Distress scale, the omega coefficient was 0.779.

## 4. Discussion

The aim of this study was to test the psychometric properties (structural validity, internal consistency reliability) of the short form of the QISO (QISO-SF), an instrument designed for assessing the nursing organizational well-being. To our knowledge, this was the first work to validate the QISO-SF, demonstrating how the psychometric properties were preserved with respect to the original version through a confirmatory statistical approach.

The instrument had good psychometric properties and could capture, through its 5 scales, 11 dimensions, and 48 items, the characteristics of the working context that determined nursing organizational well-being, using a 4-point Likert scale. An average score above 2.6 indicated the presence of the investigated phenomenon within the working context. By using the scores of the QISO-SF, it was possible to identify specific contextual variables influencing the organizational well-being.

The full version of the QISO was composed of 118 items grouped into eight scales [9]. Despite being a valuable aid for those entering into the study of organizational well-being in nursing, it required a great deal of time and attention on the part of the respondents, resulting in the quality of the collected data potentially being compromised [14]. In contrast, the QISO-SF represented a short, valid, and reliable instrument to assess nursing organizational well-being, reducing the compilation time and enhanced data quality [14].

Unlike the QISO validation study, which did not include a CFA [9], the statistical analyses conducted on the QISO-SF supports the theoretical framework, which was reflected in the CFA results conducted on each scale and dimension. The structural validity and reliability were adequate for each scale [21,25,26,27], demonstrating how the QISO-SF appears to reflect the core dimensions of the original questionnaire, while offering greater practicability.

Through its structure, the QISO-SF captured all aspects of the nursing work environment that were not considered in other instruments [37,38,39]. Some authors studied interpersonal relationships between nurses and management, without considering the importance of the work environment and its determinants in their instruments and the psychosomatic disorders that could result from it [39]. On the other hand, other authors analyzed the work context, without assessing the emotional load inherent in the nursing profession and the available resources [37]. Finally, some authors, in their instrument, did not consider burdensome aspects that were possible for nursing professionals, such as workloads, home–work balance, and interpersonal conflicts, which were important determinants of organizational well-being in work environments [38].

The characteristics of the QISO-SF were comparable with those of other instruments widely used in the literature to assess the conditions of the work environment that influenced nursing organizational well-being. The short form of the QISO maintained a good internal consistency, which could be superimposed on those obtained in short instruments for the assessment of the working environment, such as the Nursing Work Environment Questionnaire, which had reliability indices ranging from 0.645 to 0.967 [40]. The CFA, conducted on each QISO-SF scale, showed adequate fit indices, supporting the structural validity of the reduced version for all scales considered and allowing the use of individual dimensions of the same instrument. The PES-NWI, in its short version with 20 items, showed fit indices (RMSEA = 0.00, CFI = 1.00, SRMR = 0.048, and TLI = 1.004) similar to those of the dimensions of the QISO-SF, confirming the excellent construct validity for the short form of the developed questionnaire [41]. Comparable QISO-SF construct validity was also observed in relation to Shin and Lee’s (2024) [42] work environment quality questionnaire (χ^2^ (219) = 403.53, *p* < 0.001; RMSEA = 0.060 (90% CI [0.050, 0.070]); CFI = 0.910; TLI = 0.900; SRMR = 0.070).

As the QISO-SF showed good psychometric properties and retained its ability to analyze the working environment thoroughly, a significant reduction in the burden of response by nurses filling out the questionnaire was expected, resulting in a valid and reliable assessment of their organizational well-being. Monitoring the determinants of nurses’ organizational well-being through an all-encompassing instrument was crucial for healthcare organizations and nursing management, enabling them to remain competitive and ensure better care outcomes [2]. Nurses who worked in a healthy working environment enhanced their performance, reducing clinical errors and increasing patient safety [3]. The availability of a short instrument that explored the variables of the work environment in a rapid, reliable, and valid manner allowed the implementation of corrective strategies, improving clinical outcomes, organizational performance, and nurse well-being [4].

### 4.1. Limitations and Strengths

Although the study results confirmed the validity and reliability of the QISO-SF, certain limitations might be considered. Firstly, the cross-sectional nature of the primary studies did not allow researchers to consider the relationship between factor structure and time. Secondly, the selection bias inherent in cross-sectional studies, which limited data collection to a sample of participants that may not represent the study population, might be considered. However, the primary studies included an analysis of organizational well-being in different clinical settings and operational realities, partially limiting the weight of these biases on the validity of the questionnaire. Given the nature of the QISO-SF, a self-report questionnaire, the results may be subject to self-reporting bias due to social desirability or subjective interpretation of items. In addition, using different statistical analysis methods increased the methodological rigor of the study, providing a more in-depth evaluation of the instrument’s psychometric properties while giving a rigorous assessment of the reliability and validity of the QISO-SF. However, future studies should employ longitudinal validation to assess the temporal stability and test–retest reliability of the QISO-SF.

### 4.2. Managerial, Practical, and Future Research Implication

The results of the study provided implications for clinical practice and future research. Firstly, the availability of a short version of the instrument, which allowed a rapid but reliable assessment of organizational well-being, represented a real opportunity for managers of healthcare organizations. Such an instrument could be used for the initial screening of organizational conditions, allowing the early identification of possible issues and guided targeted interventions. These targeted interventions may contribute to enhanced organizational outcomes by reducing, for instance, intention to leave and absenteeism, while simultaneously fostering higher levels of job satisfaction, engagement, and organizational commitment [3,36]. Nurses who experience a sustained state of well-being tend to demonstrate improved performance [1], and such a condition may also exert a direct influence on the quality and safety of patient care, ultimately strengthening nursing-sensitive outcomes [38].

Secondly, the repeated administration of the instrument over time allowed the longitudinal monitoring of nursing organizational well-being and the evaluation of the effectiveness of any corrective interventions implemented. In this way, it will be possible to obtain an objective measurement of the impact of the strategies adopted based on the evolution of the score produced by the instrument. Finally, from a scientific point of view, longitudinal and multicenter studies exploring the trend of organizational well-being over time and analyzing the differences as a function of individual variables (e.g., stage of professional career) and contextual variables (such as organizational models, type of structure—hospital, territorial, public or private—or geographical location) are desirable. Conducting such studies will enable cross-cultural and cross-organizational validations, helping to consolidate the psychometric properties of the instrument further.

## 5. Conclusions

Nursing organizational well-being influenced the provision of high-quality care. This study provided a short version of the QISO, comprising 48 items grouped into 11 dimensions, demonstrating adequate structural validity and reliability. Therefore, QISO-SF has the potential to be an alternative and valuable instrument for assessing organizational well-being among nurses. By assessing nurses’ organizational well-being, managers can better evaluate the organizational context and identify the factors that shape everyday nursing activities. Nurses who experience a positive work environment and feel supported within their team tend to perform better. This state of well-being is directly reflected in the quality of care delivered at the bedside, leading to improved nursing-sensitive outcomes. Future studies should verify the psychometric properties of the QISO-SF in other clinical settings, including community care and international contexts.

## Figures and Tables

**Table 1 nursrep-15-00432-t001:** Characteristics of the sample (*n* = 1279).

	*n* (%)	Mean (SD), Range
Nurses’ Characteristics		
Age		34.47 (7.16), 22–62
Gender		
Male	437 (34.17)	
Female	842 (65.83)	
Not binary	-	
Civil Status		
Single	521 (40.73)	
Separated/Divorced	117 (9.15)	
Married	637 (49.80)	
Masters of science in nursing		
Yes	291 (22.75)	
No	988 (77.25)	
Years of work		11.19 (8.15), 0–39
Working hours per day		7.08 (0.78), 3–10
Monthly overtime hours per week		3.14 (5.21), 0–42
Number of work absences in the past six months		2.20 (1.02), 1–4
Working Clinical Settings		
Surgery	135 (10.55)	
Medicine	129 (10.08)	
Emergency	265 (20.71)	
Operatory room department	268 (20.95)	
Oncology	88 (6.88)	
Outpatient	113 (8.83)	
Intensive Care	251 (19.62)	
Pediatric	30 (2.35)	

Note. *n* = absolute frequency; % relative frequency; SD = standard deviation.

**Table 2 nursrep-15-00432-t002:** Factor loadings estimates from the final solutions of QISO-SF (*n* = 1147).

Scale	Dimension	Item	Factor Loading
Comfort		COM1. Cleanliness	0.732
		COM2. Lighting	0.791
		COM3. Temperature	0.662
		COM4. Quietness	0.650
		COM5. Building conditions	0.710
		COM6. Pleasantness of environment and furnishings	0.735
Organizational Context and Relational Processes	Perception of head nurses (F1)	ORG30. The head nurse’s behavior is consistent with the stated objectives.	0.758
		ORG32. The head nurse wishes to be informed about the problems and difficulties that nurses encounter in their work.	0.684
		ORG35. The head nurse involves nurses in decisions concerning their work.	0.683
		ORG36. The tasks to be performed require knowledge and skills that are not available.	0.697
	Perception of effectiveness and efficiency (F2)	ORG20. The organization’s objectives are clear and well-defined	0.758
		ORG21. There are means and resources to adequately perform one’s work	0.684
		ORG23. It is easy to obtain the information one needs	0.683
		ORG26. The organization finds adequate solutions to problems that arise	0.697
	Perception of nursing colleagues (F3)	ORG22. Nurses are generally willing to meet the needs of the organization	0.648
		ORG39. The work of each nurse represents a relevant contribution	0.668
		ORG52. Even among colleagues, people listen to and try to meet each other’s needs	0.756
	Perception of rewards (F4)	ORG41. Commitment to work and personal initiatives are appreciated (through financial rewards, public recognition, commendations, etc.)	0.857
		ORG57. Financial incentives are distributed based on performance effectiveness	0.610
	Perception of conflicts (F5)	ORG24. There are colleagues who are marginalized	0.714
		ORG34. There are nurses who act arrogantly and unfairly	0.531
		ORG44. There are people who suffer psychological abuse	0.711
Workload		CAR1. Physical fatigue	0.719
		CAR2. Mental fatigue	0.771
		CAR3. Work overload	0.823
		CAR5. Emotional overload	0.589
		ORG28. The tasks to be performed require excessive effort	0.635
		ORG48. The tasks to be performed require an excessive level of stress	0.578
Positive and Negative Indicators	Management satisfaction (F1)	POS11. Trust in the management and professional skills of the Nursing management	0.894
		POS12. Trust in the management and professional skills of the Healthcare management	0.900
		POS14. Appreciation for the human and moral qualities of the Nursing management	0.895
		POS15. Appreciation for the human and moral qualities of the Healthcare management	0.877
	Nurses caring satisfaction (F2)	POS4. Willingness to go to work	0.538
		POS5. Feeling of personal fulfillment through work	0.622
		POS16. Perception that the work of your Unit is appreciated externally	0.752
		POS17. Perception that patients are satisfied with nursing care	0.843
		POS18. Perception that patients are satisfied with medical care	0.842
	Job dissatisfaction (F3)	NEG4. Gossip	0.552
		NEG5. Resentment toward the organization	0.822
		NEG6. Aggressiveness and nervousness	0.826
		NEG7. Feeling of doing useless things	0.784
		NEG8. Feeling of having little importance in the organization	0.736
		NEG9. Feeling of working mechanically without involvement	0.664
Indicators of Psychophysical Distress		DIST1. Headache and difficulty concentrating	0.754
		DIST2. Stomachache, gastritis	0.709
		DIST4. Feeling of excessive fatigue	0.749
		DIST6. Muscle and joint pain	0.673
		DIST7. Difficulty falling asleep, insomnia	0.577

Note. When no specific dimension name is indicated, the dimension is assumed to correspond to the scale name. Item numbering reflects the sequence and coding of the original QISO: each label (e.g., ORG21, CAR21) includes a scale prefix and the corresponding original item number from the full instrument. Factor loadings come from Mplus completely standardized solutions.

**Table 3 nursrep-15-00432-t003:** Fit Indices for the QISO-SF Scales (*n* = 1147).

Scale	χ^2^ (df)	CFI	TLI	RMSEA	SRMR
Comfort	40.696 (7)	0.994	0.988	0.065	0.017
Organizational Context and Relational Processes	343.414 (94)	0.977	0.971	0.048	0.037
Workload	34.871 (7)	0.993	0.985	0.059	0.018
Positive and Negative Indicators	946.148 (85)	0.967	0.959	0.094	0.060
Indicators of psychophysical distress	36.533 (5)	0.989	0.977	0.074	0.020

Legend. χ^2^ = chi-square; df = degrees of freedom; CFI = comparative fit index; TLI = Tucker–Lewis index; RMSEA = root mean square error of approximation; SRMR = standardized root mean square residual. Note. Fit indices refer to the final re-specified CFA models.

**Table 4 nursrep-15-00432-t004:** Coefficient’s reliability of QISO-SF scale (*n* = 1147).

Scale	Dimension	McDonald ω
Comfort		0.779
Organizational Context and Relational Processes	Perception of head nurses (F1)	0.703
	Perception of effectiveness and efficiency (F2)	0.748
	Perception of nursing colleagues (F3)	0.687
	Perception of rewards (F4)	0.644
	Perception of conflicts (F5)	0.667
Workload		0.810
Positive and Negative Indicators	Management satisfaction (F1)	0.847
	Nurses caring satisfaction (F2)	0.857
	Job dissatisfaction (F3)	0.845
Indicators of Psychophysical Distress		0.779

Note. When no specific dimension name is indicated, the dimension is assumed to correspond to the scale name.

## Data Availability

The data that support the findings of this study are available on request from the corresponding author. The data are not publicly available due to privacy or ethical restrictions. Any data utilized in the submitted manuscript have been lawfully acquired in accordance with The Nagoya Protocol on Access to Genetic Resources and the Fair and Equitable Sharing of Benefits Arising from Their Utilization to the Convention on Biological Diversity.

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
