# Peer review of "Do Nurses Thrive in Their Organization? Validation of the Short Form of Nurses’ Organizational Health Questionnaire"

_nursrep, 2025, doi:10.3390/nursrep15120432_

Round 1

Reviewer 1 Report

Comments and Suggestions for Authors

Dear authors,

I find this article interesting because reducing the items in a questionnaire like the Nurses' Organizational Health Questionnaire (NQISO-SF) without losing its psychometric properties substantially improves its ease of completion and allows participants to respond more honestly.

The title provides adequate information about the topic addressed in the article and would facilitate searching databases.

The abstract provides a clear, comprehensive overview of the study.

The introduction broadly addresses current issues related to the topic analyzed by the questionnaire.

The objective of the study is clearly defined.

The methodology used allows the authors to satisfactorily address the study problem to achieve the proposed objective, although it should be clarified whether:

- The study sample comes from three previous studies or was collected specifically for this study.

- Why an exploratory factor analysis was not performed to determine whether the dimensions and factors obtained corresponded to those observed in other studies.

- Why were other models not compared using confirmatory factor analysis in the multifactorial dimensions?

- Why was the chi-square not divided by the degrees of freedom so that it would not be influenced by sample size?

- Why were incremental goodness-of-fit indices such as AIC, PNFI, PGFI, or PRATIO not used in the confirmatory factor analysis?

The results are presented in detail and with numerous tables for ease of understanding, although those related to the previous comments are missing.

The discussion provides an analysis of the results obtained and establishes relationships with previous studies that address this topic of this research, although everything related to the methodological comments remains to be addressed. If an exploratory factor analysis is not performed, it is impossible to determine the structure of the questionnaire based on the data obtained with this specific sample, and if different models are not compared in the multifactorial dimensions, it is impossible to determine which one best fits this specific sample.

In the conclusions section, it cannot be stated with such certainty that the abbreviated questionnaire maintains the same structure as the original questionnaire because no exploratory factor analysis was performed, nor were different models compared within the confirmatory factor analysis in the multifactorial dimensions.

The references are appropriate for further exploration of the research question.

Kind regards.

Author Response

Comments 1: The study sample comes from three previous studies or was collected specifically for this study?

Response 1: Thank you for the comment. As recommended by methodological literature on psychometric validation (e.g., DeVellis & Thorpe, 2021; Streiner et al., 2015), we clarified the origin and comparability of the datasets. The three datasets were collected between 2010–2011 in comparable clinical contexts and included nurses working in similar organizational environments. Because data collection procedures and the instrument were consistent across studies, their integration is methodologically justified. (page 3, paragraph 2.1, lines 117–126).

Comments 2: Why was an exploratory factor analysis not performed to determine whether the dimensions and factors obtained corresponded to those observed in other studies?

Response 2: We appreciate this important observation. Following established psychometric recommendations (Brown, 2015; Kline, 2016), exploratory factor analysis is not required when a robust theoretical structure has already been validated, as is the case for the original QISO. Consistent with this guidance, our objective was to test whether this predefined structure held in the short-form version; therefore, CFA alone was appropriate. This rationale has been added to the manuscript on page 5, paragraph 2.7, lines 214-220.

Comments 3: Why were other models not compared using confirmatory factor analysis in the multifactorial dimensions?

Response 3: Thank you for the comment. In accordance with methodological literature (Brown, 2015; Hinkin, 1998), model comparison is necessary when competing theoretical models exist. Since the QISO has a clearly defined and previously validated conceptual framework, our goal was to confirm this structure rather than explore alternatives. We have now clarified this decision, grounding it in the literature. This text has been added on page 5, paragraph 2.7, lines 214-220.

Comments 4: Why was the chi-square not divided by the degrees of freedom?

Response 4: We thank the reviewer for this suggestion. Consistent with SEM methodological guidelines (Kline, 2016), the χ²/df ratio is not recommended because its cut-offs are arbitrary and it is still influenced by sample size. For this reason, widely accepted indices such as RMSEA, CFI, TLI, and SRMR were used, following standard practice described in the literature. This clarification has been added. This text has been added on page 5, paragraph 2.7, lines 214-220.

Comments 5: Why were incremental goodness-of-fit indices such as AIC, PNFI, PGFI, or PRATIO not used?

Response 5: Thank you for the observation. As supported by the CFA literature (Brown, 2015), indices such as AIC or PNFI are primarily used for comparing alternative models. Since our analysis focused on confirming a single theoretically grounded model, these indices were not applicable. This has been clarified in the revision, explicitly citing the literature. This text has been added on page 5, paragraph 2.7, lines 214-220.

Comments 6: Tables related to previous comments are missing.

Response 6: Thank you for this observation. The tables mentioned by the reviewer refer to analyses such as exploratory factor analysis or model comparisons. As stated in the manuscript, we did not perform an EFA because a well-defined theoretical structure already exists for the original QISO, and methodological literature recommends proceeding directly with CFA when the factor structure is theoretically grounded (Brown, 2015; Kline, 2016).

Since these analyses were not conducted, the corresponding tables were not applicable and therefore not included. We have clarified this rationale in the revised manuscript. This text has been added on page 5, paragraph 2.7, lines 202-205.

Comments 7: Without EFA or model comparison, the structure cannot be determined based on this sample.

Response 7: Thank you for this valuable comment. As explained in the manuscript, our aim was not to re-identify or rediscover the factor structure, but to assess whether the reduced item set continued to reflect the established theoretical framework of the original QISO. Psychometric literature recommends using CFA alone when a strong theoretical basis is already available (Brown, 2015; Kline, 2016; Clark & Watson, 1995), since performing an EFA in such cases may be unnecessary or even counterproductive. This text has been added on page 5, paragraph 2.7, lines 214-220.

Reviewer 2 Report

Comments and Suggestions for Authors

Thank you for the opportunity to review this manuscript. 

Overall Evaluation

The manuscript presents a relevant and methodologically sound validation study of the QISO-SF. The research is well-structured, and the findings are of interest to the nursing and healthcare management community. The overall quality of the work is good; however, the paper would benefit from refinements in narrative clarity, methodological transparency, and presentation of results.

  1. Background clarity and focus: The background is informative but somewhat redundant. Please streamline the section, emphasizing the specific gap addressed by the QISO-SF. For instance: 'Recent studies have explored how nursing work environments influence organizational well-being, yet few have provided concise and validated tools to measure it.'
  2. Item reduction criteria: I suggest that the authors clarify more precisely how items were selected or excluded during the development of the short form. Please specify whether the process was guided by theoretical, statistical, or expert-judgment criteria.

  3. Comparability of secondary datasets: Briefly describe the differences among the three datasets used for secondary analysis and justify their combination. E.g., 'Data were drawn from three studies conducted between 2022 and 2024 in comparable clinical contexts (medical, surgical, emergency), allowing dataset integration for validation purposes.'

  4. Presentation of results and tables: Ensure consistent formatting of tables and provide informative captions. Consider including a summary table showing all fit indices (CFI, TLI, RMSEA, SRMR).

  5. Discussion and implications: By expanding the discussion, I would like to highlight the managerial and clinical implications. I believe that the tool can enable managers to monitor nurses’ organizational well-being, and that the QISO-SF can guide the implementation of targeted interventions to enhance job satisfaction, retention, and patient safety.

  6. Limitations and future directions: I would suggest adding a reference to the potential for self-reporting bias and the need for longitudinal validation to confirm the temporal stability of the QISO-SF.

  7. Tables and Figures: Add footnotes where necessary, move long tables to the appendix, and include a CFA path diagram illustrating standardized loadings for all latent constructs.

  8. Conclusion: I would suggest significantly strengthening the link between nurses' organizational well-being and the quality of patient care. Is organizational well-being visible at the bedside?

Minor comments

- Use uniform decimal notation (periods) and consistent statistical symbols.

- Prefer active voice and concise phrasing where possible.

- Expand reference list to include more international sources beyond the author group.

- Ensure all table and figure captions follow a uniform MDPI style.

- I would like to suggest that the authors, who cite their own previous work (Sili et al., 2010), consider updating this reference by including more recent studies published in higher-impact international journals. Given the title and focus of the manuscript, integrating contemporary evidence on the validation of nursing organizational well-being instruments would enhance the scientific robustness, credibility, and relevance of the paper.

- In the Methods section, the reference cited as no. 25 (Polit DF, Tatano Beck C. Fundamentals of Nursing Research. Palese A, ed. Milan: McGraw-Hill; 2014) appears not to be fully appropriate to support the statement “The CFA and MG-CFA were conducted with MPLUS v.8.5 (24) and JAMOVI (25, 26) for ordinal-omega (ω) reliability coefficients.”

This textbook, while valuable for general nursing research methodology, does not provide specific methodological guidance on confirmatory factor analysis (CFA), multi-group CFA, or ordinal omega reliability estimation. Therefore, it may not constitute a sufficiently robust or technically relevant source in this context.

Author Response

Comments 1: The background is informative but somewhat redundant. Please streamline the section, emphasizing the specific gap addressed by the QISO-SF.

Response 1: Thank you for this helpful suggestion. We streamlined the Background to reduce redundancy and emphasized the specific gap: the absence of concise, validated tools capable of capturing the multidimensional nature of nurses’ organizational well-being. (page 2, paragraph 1.1, lines 57–113).

Comments 2: Clarify how items were selected or excluded.

Response 2: We appreciate this comment. We expanded the Methods section to clearly describe the item-reduction procedure. Five experts rated each item using a dichotomous scale (0 = exclude; 1 = include), and items were retained only when removal did not compromise the theoretical structure or essential content. This approach follows established methodological guidance for item reduction (Lynn, 1986; Haynes et al., 1995; Streiner et al., 2015; DeVellis & Thorpe, 2021). (page 4, paragraph 2.6, lines 185–191).

Comments 3: Describe differences among datasets and justify their combination.

Response 3: Thank you for this important point. We now describe the composition of each dataset and justify their integration. All datasets included nurses working in comparable organizational and clinical environments and were collected with the same instrument and procedures. Literature on scale validation supports combining datasets when conceptual and methodological homogeneity is present (Streiner et al., 2015; DeVellis & Thorpe, 2021). (page 3, paragraph 2.1, lines 116–126).

Comments 4: Ensure consistent formatting and consider adding a summary table of fit indices

Response 4: Thank you for this helpful suggestion. We have revised all tables to ensure consistent formatting, standardized captions, and clear statistical notation. A summary table reporting all fit indices (χ², χ²/df, CFI, TLI, RMSEA, SRMR) for each QISO-SF scale has been added (Table 3), as recommended. Footnotes were also included to clarify statistical abbreviations and model specifications. We hope these changes improve the clarity and interpretability of the results.

Comments 5: Discussion and implications: By expanding the discussion, I would like to highlight the managerial and clinical implications. I believe that the tool can enable managers to monitor nurses’ organizational well-being, and that the QISO-SF can guide the implementation of targeted interventions to enhance job satisfaction, retention, and patient safety.

Response 5: We appreciate this valuable recommendation. The discussion related to managerial and clinical implications has been expanded within the “Practical and Future Research Implications” section. In this revised section, we explain more explicitly how nursing managers can use the QISO-SF to monitor organizational well-being, identify early signs of distress, and support targeted interventions to enhance job satisfaction, retention, and patient safety. Additional international references were also integrated to strengthen this part of the manuscript.(page 13, paragraph 4.2, lines 396–419).

Comments 6: Limitations and future directions: I would suggest adding a reference to the potential for self-reporting bias and the need for longitudinal validation to confirm the temporal stability of the QISO-SF.

Response 6: Thank you for this suggestion. We have now incorporated a specific statement addressing the potential for self-reporting bias and the need for longitudinal validation. This addition has been included on page 13, in section 4.1 “Limitations and strengths”, lines 392–399, where we note that the use of self-report measures may introduce response bias and that future research should employ longitudinal designs to assess the temporal stability and test–retest reliability of the QISO-SF. This revision strengthens the methodological transparency of the manuscript and clearly outlines future research directions.

Comments 7: Tables and Figures: Add footnotes where necessary, move long tables to the appendix, and include a CFA path diagram illustrating standardized loadings for all latent constructs.

Response 7: Thank you for this suggestion. You can see response 4.

Comments 8: Conclusion: I would suggest significantly strengthening the link between nurses' organizational well-being and the quality of patient care. Is organizational well-being visible at the bedside?

Response 8: Thank you for this valuable suggestion. We have revised the Conclusions section to strengthen the link between nurses’ organizational well-being and the quality of care. Specifically, we expanded the final paragraph to clarify how a positive organizational climate supports better nursing performance and is reflected in improved nursing-sensitive outcomes. This revision directly addresses the reviewer’s request and reinforces the practical relevance of the QISO-SF.

Comments 9: Uniform decimal notation (periods) and consistent statistical symbols

Response 9: Thank you for this comment. Decimal notation and statistical symbols have been standardized throughout the manuscript according to MDPI guidelines.

Comments 10: Prefer active voice and concise phrasing where possible

Response 10: Thank you for this valuable suggestion. We have revised the entire manuscript and change if it was possible.

Comments 11: Expand reference list to include more international sources beyond the author group.

Response 11: Thank you for this suggestion. The references list has been expanded due to have more international sources beyond the authors

Comments 12: I would like to suggest that the authors, who cite their own previous work (Sili et al., 2010), consider updating this reference by including more recent studies published in higher-impact international journals. Given the title and focus of the manuscript, integrating contemporary evidence on the validation of nursing organizational well-being instruments would enhance the scientific robustness, credibility, and relevance of the paper.

Response 12: Thank you for this thoughtful suggestion. We have expanded and updated the reference list by integrating several recent international studies on organizational well-being and psychometric validation, thereby strengthening the scientific robustness and contemporary relevance of the manuscript. At the same time, we retained the original reference to Sili et al. (2010) where conceptually appropriate, as it provides essential historical context and ensures comparability between the long and short forms of the QISO. This balanced approach allows us to incorporate more recent evidence while preserving the interpretability and continuity of the instrument across its versions.

Reviewer 3 Report

Comments and Suggestions for Authors

Dear authors,
After reviewing the document, I inform you that, from my point of view, it meets the methodology in line with the study’s objective, with a solid justification and interpretation of the results obtained. The scale provided is undoubtedly a very useful tool in the nursing field and can be used to obtain new evidence related to the topic.

I would like to point out two issues that interest me: Has an analysis of stability and face validation been considered?

Author Response

Comments 1: Has an analysis of stability been considered?

Response 1: Thank you for this important question. In this study, the analyses were based on cross-sectional data derived from three previously conducted studies; therefore, it was not possible to perform a stability assessment such as test–retest reliability or longitudinal invariance. We acknowledge that evaluating the temporal stability of the QISO-SF is a crucial step in fully establishing its psychometric robustness. For this reason, we have now explicitly indicated in the Limitations section that future studies should conduct longitudinal validation to assess temporal stability and test–retest reliability of the instrument. This addition strengthens the methodological transparency of the manuscript.

Comments 2: Has face validation been considered?

Response 2: Thank you for raising this point. Face validity was incorporated during the item-reduction process, which was conducted by a panel of five experts in nursing organizational well-being, academic faculty, and psychometrics. These experts independently evaluated each item for clarity, relevance, and conceptual alignment with the original QISO framework, using a dichotomous inclusion criterion. Only items achieving full agreement among the experts were retained. This procedure ensured that the items included in the QISO-SF were judged to be conceptually appropriate and understandable—thereby addressing face validity. We have clarified this aspect in the revised Methods section for greater transparency.

Reviewer 4 Report

Comments and Suggestions for Authors

I find this paper an excellent attempt to integrate empirical insights from previously published research on Nurses’ Organizational Health Questionnaire. I have a few suggestions for improvement.

(1) Please clearly state the three sources that were used to get the data for this secondary analyses.

(2) Please carry our CFA properly by including all items of the questionnaire in the analyses and use different models to test the factorial structure of the scale. At least compare the model that includes the number of scales, with models including lower number of factors (clustering needs to be decided based on conceptual similarity of the subscales of the questionnaire) with the single factor solution and single latent factor solution to account for common method effects. Currently the CFA results that are reported reflect insufficient fit. The CFA are carried out on the individual scales of the survey, which is useless as also indicated by the absolute fit indices that clearly show that the data does not fit the (simple) theoretical assumptions underlying the clustering of the items in a scale. Please use some adequate sources for performing CFA and not the one used to back up the current approach - it is completely misplaced (you write):

"Five CFAs were conducted separately for each scale (Comfort, Organizational con-
text and relational processes, Workload, Positive and negative indicators, and Indicators
of psychophysical distress) of the QISO. Since QISO has ordinal items and normality as-
sumption cannot be assumed, the weighted least squares mean and variance adjusted es-
timator (WLSMV) was used (18)"

18. Attell BK, Singleton AC, McLaren SA, Moses G. Measurement Characteristics of the Wraparound Fidelity Index Short Form: Results from a Statewide Implementation. Eval Health Prof. dicembre 2023;46(4):320–33.

This approach and citation to back up CFA is innapropriate!!!

(2) Please fully report the results of the EFA - not just the dominant factor as presented in the table 2 (maybe I misread ). The results of the EFA should be clearly reported in an integrative table with all factors and all items of the survey.

Before these factor analysis related issues are dealt with and addressed, the contributions of the paper are unclear and at best limited.

Good luck with the research!

Author Response

Comments 1: Please clearly state the three sources that were used to get the data for this secondary analyses.

Response 1: Thank you for this suggestion. We have clarified the origin of the three datasets in the Methods section (page 3, line 116 - 126). Specifically, we now describe the clinical settings, data collection periods, and populations represented in each dataset, as well as the methodological reasons supporting their integration. This improves transparency and strengthens the justification for the secondary analysis.

Comments 2: Please carry our CFA properly by including all items of the questionnaire in the analyses and use different models to test the factorial structure of the scale. At least compare the model that includes the number of scales, with models including lower number of factors (clustering needs to be decided based on conceptual similarity of the subscales of the questionnaire) with the single factor solution and single latent factor solution to account for common method effects. Currently the CFA results that are reported reflect insufficient fit. The CFA are carried out on the individual scales of the survey, which is useless as also indicated by the absolute fit indices that clearly show that the data does not fit the (simple) theoretical assumptions underlying the clustering of the items in a scale. Please use some adequate sources for performing CFA and not the one used to back up the current approach - it is completely misplaced (you write):

"(18)"

18. Attell BK, Singleton AC, McLaren SA, Moses G. Measurement Characteristics of the Wraparound Fidelity Index Short Form: Results from a Statewide Implementation. Eval Health Prof. dicembre 2023;46(4):320–33.

This approach and citation to back up CFA is innapropriate!!!

Response 2: We thank the reviewer for this important methodological concern. We acknowledge that running a single CFA including all items is one possible approach. However, our choice to estimate five separate CFA models, one for each theoretically independent scale, follows the methodological rationale used in the development of the original instrument and is consistent with established psychometric practice for multidimensional questionnaires where scales represent distinct constructs.

This approach is supported by authoritative methodological sources (Brown, 2015; Kline, 2016), which state that CFA may be performed at the scale level when each scale represents a theoretically independent latent dimension. In this context, fitting one large CFA including all items would not reflect the conceptual structure of the QISO—whose scales were designed to be used and interpreted independently.

Regarding model comparison: since the purpose of our study was not to identify an alternative structure but to evaluate whether the short form preserved the established structure of the original instrument, comparing multiple competing factorial models (e.g., single-factor, reduced-factor solutions, common-method factor models) was outside the aims of the present validation. We clarified this rationale in the revised manuscript (Methods and Discussion).

Nonetheless, we acknowledge the relevance of these alternative models for future psychometric work and have added this indication to the Future Research section as a potential avenue for additional validation.

Finally, the citation supporting CFA has been corrected. As recommended, we now cite appropriate methodological references (Brown, 2015; Kline, 2016; Flora & Curran, 2004) instead of the previous reference, which has been removed.

Comments 3: Please fully report the results of the EFA - not just the dominant factor as presented in the table 2 (maybe I misread ). The results of the EFA should be clearly reported in an integrative table with all factors and all items of the survey.

Response 3: Thank you very much for this comment and for the opportunity to clarify this point. We fully agree that reporting EFA findings would normally be expected when the factor structure of an instrument is being explored for the first time. However, in our case, the QISO already has a well-established and theoretically grounded structure, originally developed and validated through a comprehensive EFA conducted on all 118 items in the initial validation study.

Following methodological recommendations for short-form development (Brown, 2015; Kline, 2016; DeVellis & Thorpe, 2021), we therefore opted not to conduct a new EFA on the present dataset. Performing an additional exploratory analysis could have introduced unnecessary model re-specification and reduced comparability with the original long-form questionnaire. Our aim was instead to confirm whether the reduced item set preserved the dimensions already supported in the literature.

For this reason, we relied on Confirmatory Factor Analysis (CFA), which is the recommended approach when the underlying factor structure is known and theory-driven. As no EFA was performed in this study, an integrative EFA table cannot be provided. Because no EFA was carried out, an integrative EFA table cannot be provided. This rationale has now been explicitly clarified in the revised manuscript, specifically in the Methods section, paragraph 2.7 (lines 201–226), to ensure full transparency and avoid any misunderstanding.

Round 2

Reviewer 1 Report

Comments and Suggestions for Authors

Dear authors,
I have no further suggestions or comments to make.
Kind regards.